# Synthesis of Novel Indole Schiff Base Compounds and Their Antifungal Activities

**DOI:** 10.3390/molecules27206858

**Published:** 2022-10-13

**Authors:** Caixia Wang, Liangxin Fan, Zhenliang Pan, Sufang Fan, Lijun Shi, Xu Li, Jinfang Zhao, Lulu Wu, Guoyu Yang, Cuilian Xu

**Affiliations:** College of Sciences, Henan Agricultural University, Zhengzhou 450002, China

**Keywords:** antifungal activities, synthesis, indole Schiff base derivatives, 1,3,4-thiadiazole, *F. graminearum*, *F. oxysporum*, F. *moniliforme*, C. lunata, *P. p.* var. *nicotianae*

## Abstract

A series of novel indole Schiff base derivatives (**2a**–**2t**) containing a 1,3,4-thiadiazole scaffold modified with a thioether group were synthesized, and their structures were confirmed using FT-IR, ^1^H NMR, ^13^C NMR, and HR-MS. In addition, the antifungal activity of synthesized indole derivatives was investigated against *Fusarium graminearum* (*F. graminearum*), *Fusarium oxysporum* (*F. oxysporum)*, *Fusarium*
*moniliforme* (*F.*
*moniliforme*), *Curvularia lunata (C. lunata*), and *Phytophthora parasitica* var. *nicotiana* (*P. p.* var. *nicotianae*) using the mycelium growth rate method. Among the synthesized indole derivatives, compound **2j** showed the highest inhibition rates of 100%, 95.7%, 89%, and 76.5% at a concentration of 500 μg/mL against *F. graminearum*, *F. oxysporum*, *F.*
*moniliforme*, and *P. p.* var. *nicotianae*, respectively. Similarly, compounds **2j** and **2q** exhibited higher inhibition rates of 81.9% and 83.7% at a concentration of 500 μg/mL against *C. lunata*. In addition, compound **2j** has been recognized as a potential compound for further investigation in the field of fungicides.

## 1. Introduction

Food crop diseases caused by fungi have become one of the concerns in the global agricultural sector [1]. Fungal diseases directly cause a reduction in crop yield and quality, which results in a huge economic loss for farmers worldwide [1,2]. Furthermore, some pathogenic fungi can secrete toxins and metabolites that are harmful to humans and livestock [3,4,5,6]. For example, *F. oxysporum* is a soil-borne fungal pathogen widely distributed throughout the world that can infect more than 100 valuable crops by causing blight and root rot, seriously affecting plant growth, yield, and quality [7,8,9,10,11,12,13]. Similarly, *F. graminearum* is responsible for fusarium head blight (FHB) disease in barley, rice, and oat, and stem rot and spike rot in maize, which severely affects the production of these crops on a global scale [14,15,16,17,18]. Meanwhile, mycotoxins such as trichothecenes and zearalenone produced by *F. graminis* are harmful to humans and livestock [19]. The use of fungicides is the most common and well-known method for controlling these fungal diseases. However, the excessive or improper use of antifungal agents leads to an increase in the resistance of fungi to fungicides. Thus, the discovery of new antifungal compounds with a new mechanism of action is of great significance for future development in agriculture.

In recent years, heterocyclic pesticides have become the mainstream of pesticide research because of their flexible structure, low toxicity, and high activity. Indole is an important nitrogen-containing heterocyclic compound. Indole and indole derivatives have a broad spectrum of biological activities such as antifungal [20,21,22,23,24,25,26], antibacterial [27,28,29], antimycobacterial [30], antitubercular [31,32,33], antioxidant [34], antimalarial [35,36,37], antiviral [38,39,40,41], anti-leishmanial [42,43], anti-inflammatory [44], and anti-tumor [45,46,47] activities. The design and synthesis of new indole derivatives with excellent biological activity is one of the emerging fields in pharmaceutical chemistry. There are various indole-based drugs available for the treatment of human fatal diseases (I–VI in Figure 1). Furthermore, some indole derivatives containing coumarin [48], thiofuran [49], oxazole [50], and imidazole [51] at the 3-position of the indole ring were found to exhibit obvious fungicidal activity (VII–X in Figure 1). However, the usage of commercial indole-based pesticides for the treatment of plant fungal diseases has not yet been explored.

1,3,4-thiadiazole derivatives were widely used as pesticides in agrochemical chemistry and studied for years due to their excellent biological activities, including antifungal [52,53], insecticidal [54], acaricidal [54], antibacterial [55], and herbicidal activities [56]. Heterocyclic thioether compounds also possess high antifungal activities [57], and the thioether-bound 1,3,4-thiadiazole scaffold is an important pharmacophore [58]. Some thioether-bound 1,3,4-thiadiazole derivatives such as bismerthiazol and 2,5-dimercapto-1,3,4-thiadiazole zinc salts (I, II in Figure 2) have been used as commercial fungicides for plant fungal diseases. Schiff base, a class of compounds with imine groups (-CH=N-), is a common pharmacological group in many compounds, which has a wide range of biological activities [59,60]. The introduction of Schiff bases into 1,3,4-thiadiazoleis is interesting to study, and the 1,3,4-thiadiazole Schiff base derivatives also have biological activities [61,62]. For example, compound V in Figure 2 were found to exhibit obvious fungicidal activity [63].

In this research, our aim is to find new antifungal compounds to control fungal diseases from farmland. Based on the different advantages of indoles, thiadiazoles, thioethers and Schiff bases, and in continuation of our long-term research on the heterocyclic derivatives such as 1,3,4-thiadiazole [64], coumarin [65,66,67,68,69,70,71], indole [72], and chitosan [73] as agricultural antifungal agents, herein, we designed and synthesized a series of target compounds **2a**–**2t** containing indole, thioether-modified 1,3,4-thiadiazole, and imine. These compounds have obvious inhibitory activities against plant pathogenic fungi, which have not been reported in the literature at home and abroad. The structure–activity relationship of the new derivatives against fungi was determined. This structure–activity relationship lays a foundation for the research and development of drugs to control plant fungal diseases in the future. According to the preliminary inhibition experiments results, compound **2j** had been recognized as a potential compound for further investigation in the field of fungicides. The design of target compounds is shown in Figure 1.

## 2. Results and Discussion

### 2.1. Synthesis

The synthetic pathway used in the preparation of novel indole derivatives **2a**–**2t** containing thioether-modified 1,3,4-thiadiazole is shown in Figure 2. The (1*H*)-indole-3-formaldehyde was condensed with 2-amino-5-alkylthio-1,3,4-thiadiazole in ethanol or 1,4-dioxane solvent in the presence of a CH_3_COOH catalyst to obtain the target compounds. The progress of the reaction was monitored using HPLC and TLC. The reaction progress monitoring revealed that it took approximately 4–6 h to completely consume the 2-amino-5-alkylthio-1,3,4-thiadiazole, and the target compounds **2a**–**2i** can be obtained with a yield range of 62–94% after refluxing at 80 °C with ethanol as the solvent. However, compounds **2k**–**2t** showed low yields or no product under the same conditions. After refluxing at 100 °C with 1,4-Dioxane instead of ethanol as solvent, the result was higher yields of compounds **2k**–**2t**. Conclusively, the formation of compounds **2k**–**2t** required higher temperatures compared to the formation of compounds **2a**–**2j**.

The structures of the synthesized compounds **2a**–**2t** were confirmed using different spectroscopic techniques, such as FT-IR, ^1^H NMR, ^13^C NMR, and HR-MS analyses. The FT-IR spectra of the synthesized compounds **2a**–**2t** showed one or two separate absorption bands in the 3267–3506 cm^−1^ region, which corresponds to the N–H stretching of the indole ring. The peaks corresponding to aromatic =C–H and C=N stretching bands were identified in the 3040–3097 cm^−1^ and 1605–1698 cm^−1^ regions, respectively. The peak was observed at 1035–1087 cm^−1^ and corresponds to the thioether bond C–S–C stretching. The ^1^H NMR spectra of compounds **2a**–**2t** showed the pyrrole N–H protons of the indole moiety as one singlet in the δ 11.18–12.34 ppm region and the C–H protons of the imine group as one singlet in the δ 8.90–10.68 ppm region. The thioether (SCH_2_) C–H proton signals of compounds **2j** and **2k** were observed at δ 3.75 and 4.37 ppm, respectively, as a doublet due to the ortho coupling with the ethylene C–H. In the other compounds, the thioether (SCH_2_) C–H protons were found as one singlet in the δ 4.29–4.83 ppm region. The ^13^C NMR spectrum showed the resonances of C=N, S–C, 1,3,4-thiadiazole C2, and 1,3,4-thiadiazole C5 through the signals at δ 152.80–165.56, 19.02–38.09, 170.49–181.65, and 161.88–170.08 ppm, respectively. The HR-MS of compounds **2a**–**2t** was conducted using the electrospray ionization method (ESI). In the HR-MS spectra of compounds **2a**–**2t**, [M + H^+^], [M + Na^+^] or [M–H^+^] peaks were observed, which confirmed their precise molecular weights.

The synthesized compounds **2a**–**2t** had moderate solubility in ethanol and methanol, and good solubility in DMF, DMSO, acetone, and chloroform. The synthetic molecules are stable in any of the above solvents.

### 2.2. In Vitro Antifungal Activity

The results of the preliminary inhibition experiments of the target compounds **2a**–**2t** against *F. graminearum*, *F. oxysporum*, *F.*
*moniliforme*, *C. lunata*, and *P. p.* var. *nicotianae* are shown in Figure 3, Figure 4, Figure 5, Figure 6 and Figure 7, respectively. Photos of some of the compounds in the fungal inhibition experiment are in the Appendix A.

From the experimental results, we found that the target compounds **2a**‒**2t** has different inhibitory activities against the experimental fungi. For example, at the concentration of 500 µg/mL, the inhibitory rate of the target compound **2a**–**2t** against *F. graminearum* was within the range of 36.8–100% (in Figure 3). Among the tested compounds, compounds **2i**, **2j**, **2m**, **2n**, **2o**, **2p**, **2q**, and **2r** exhibited higher inhibition rates than the control reagent triadimefon (inhibition index: 47.6%). At the same concentration, the inhibitory rate of compounds **2a**–**2t** against *F. oxysporum* was in the range of 45.2–95.7% (in Figure 4), which was higher than that of the control drug triadimefon (the inhibitory rate of 45.2%). Some compounds, such as compound **2j** and compound **2q**, showed a broad spectrum of good antifungal activity. The inhibition rates of compound **2j** against *F. Graminearum*, *F. oxysporum*, *F. Moniliforme*, *C. lunata*, and *P. p. var. nicotianae* were 100%, 95.7%, 91.2%, 81.9%, and 82.1%, respectively. Compound 2q showed better inhibitory activity against *F. graminearum* and *C. lunata* with inhibition rates of 76.5% and 83.7%, respectively.

The structure–activity relationship indicated that different substituents attached to the benzene ring of the target compounds would have obvious effects on the inhibitory activity of the experimental fungi. The introduction of electron-withdrawing groups such as –NO_2_, –CF_3_, –F, –Cl on the benzene ring resulted in an increase in the antifungal activity of compounds such as **2j**, **2i**, **2k**, **2e**, **2p**, **2q**, **2r**, **2s**, and **2t**, compared to compound **2a**. The higher antifungal activity of those compounds may be due to the electron-withdrawing group on the benzene ring, which decreases the electron cloud density and results in an increase in the accessibility of the target molecules toward the fungicide cell. In addition, different positions of the same substituents have different effects on the inhibitory activities of different fungi. For example, when –CF_3_ is in different positions (ortho:**2t**, meta:**2p** and para:**2s**), it has little effect on the inhibition rate of compound against *F. Graminearum*, *F. Oxysporum*, *F. Moniliforme* and *C. Lunata*, but has a great effect on the inhibition rate of compound against *P. p. var. nicotianae*. The inhibition rates of meta compound (**2p**) and ortho compound (**2t**) against *P. p. var. nicotianae* were 53.4% and 33.9%, respectively. The inhibition rates of 3, 5-di-substituted –CF_3_ compound (**2q**) against *F. Graminearum* and *C. lunata* were 76.5% and 83.7%, respectively, which were higher than that of mono-substituted –CF_3_ compounds (**2p**, **2s** and **2t**). However, the inhibition rates of 3, 5-di-substituted –CF_3_ compound (**2q**) against the other three fungi were almost the same as that of mono-substituted –CF_3_ compounds (**2p**, **2s**, **2t**). The compounds with different substituted pyridine positions had different inhibitory activities against fungi. The inhibition rates of 4-position pyridine compound (**2m**) against *F. Graminearum*, *F. Moniliforme*, *C. Lunata*, and *P. p. var. nicotianae* were higher than that of 2-position and 3-position pyridine compounds (**2b** and **2c**). However, the inhibition rate of 2-position pyridine compound (**2b**) against *F. oxysporum* was higher than that of 3-position and 4-position pyridine compounds (**2c** and **2m**).

## 3. Materials and Methods

### 3.1. Chemicals and Instruments

All reagents and chemicals were procured from a commercial supplier (Shanghai Aladdin Reagent Co., Ltd., Shanghai, China) and used as received. The method described in the literature was used to synthesize the intermediate **1** (**1a**–**1t**, 2-ammino-5-alkylthio-1,3,4-thiadiazoles) [64]. Five crop-threatening pathogenic fungi (*F. graminearum, F. oxysporum, F.*
*moniliforme, C. lunata,* and *P. p.* var. *nicotianae*) were obtained from the College of Plant Protection of Henan Agricultural University.

The Fourier transformed infrared (FT-IR) spectra were recorded using a Thermo Scientific Nicolet IS10 FT-IR spectrometer (Nicolet Technologies Co., Madison, America) and the frequencies were given in cm^–1^. The proton nuclear magnetic resonance (^1^H NMR) and carbon nuclear magnetic resonance (^13^C NMR) spectra were obtained using a Bruker DPX-400 spectrometer (Brucker Technologies Co., Karlsruhe, German) in acetone or dimethyl sulfoxide (DMSO) solvent with tetramethylsilane (TMS) as an internal standard. A thin-layer chromatography (TLC) was performed on silica gel 60 F254 (Shanxi ersai biotechnology Co., Ltd., Xian, China). A high-performance liquid chromatography (HPLC) from Thermo Fisher Science and Technology Ltd. with C18 chromatographic column was used in the process of the reaction. The high resolution-mass spectroscopy (HR-MS) was performed using an Ultimate 3000RE-Q-Exactive^TM^ Orbitrap, Thermo Fisher-ESI instrument (Thermo Fisher Technologies Co., Waltham, America). Melting points were determined using a Taike X-4 melting point apparatus. The reaction yields, except for compound **2a**, were not optimized.

### 3.2. General Procedure for the Preparation of Compounds ***2a***–***2t***

A total of 3.6 mmol of 3-indoxformaldehyde and 3 mmol of the intermediate **1a** (2-amino-5-S-benzyl-1,3,4-thiadiazole) were taken in the round bottom flask and dissolved in ethanol, and then a few drops of acetic acid were added as a catalyst. The resulting mixture was refluxed for 5 h at 80 °C. Once the reaction was completed according to thin layer chromatography (TLC) or high-performance liquid chromatography (HPLC), the reaction solution was cooled and then filtered using vacuum filtration to obtain the crude product. The crude product was then purified using ethanol recrystallization to obtain the desired product **2a**. The preparation method for compounds **2b**–**2t** was the same as for compound **2a**.

### 3.3. Spectral Data

(E)-N-(5-(benzylthio)-1,3,4-thiadiazol-2-yl)-1-(1H-indol-3-yl) methanimine (**2a**)

Orange yellow crystal; M. p. 200.5–201.4 °C; yield 72%; IR (ν, cm^–1^ KBr): 3506 (N–H), 3069 (Ar–H), 1630 (C=N), 1524, 1513, 1402, 1336, 1204 (thiadiazole ring), 1042 (C–S–C); ^1^H NMR (400 MHz, DMSO, *d_6_*, δ, ppm): 12.31 (s, 1H, N–H), 8.92 (s, 1H, HC=N), 8.30 (d, *J* = 8.0 Hz, 2H, Ar–H), 7.55 (d, *J* = 8.0 Hz, 1H, Ar–H), 7.46 (d, *J* = 8.0 Hz, 2H, Ar–H), 7.35 (t, *J* = 8.0 Hz, 2H, Ar–H), 7.29 (t, *J* = 8.0 Hz, 3H, Ar–H), 4.56 (s, 2H, SCH_2_); ^13^C NMR (101 MHz, DMSO *d_6_*, δ, ppm): 176.51, 163.41 (thiadiazole ring), 160.51 (C=N), 139.14, 138.02, 137.05, 129.56, 129.04, 128.10, 124.93, 124.23, 122.67, 122.38, 114.76, 113.09, 37.83 (SCH_2_); HR-MS (ESI): calcd. for C_18_H_14_N_4_S_2_: [M + Na^+^] 373.0558; found: 373.0559.

(E)-N-(5-((pyridin-2-ylmethyl)thiol)-1,3,4-thiadiazol-2-yl)-1-(1H-indol-3-yl)methanimine (**2b**)

Yellow needle-shaped crystal; M. p. 209.1–210.5 °C; yield 81%; IR (ν, cm^–1^ KBr): 3442 (N–H), 3091(Ar–H), 1619 (C=N), 1596, 1573, 1478, 1429, 1374, 1245 (thiadiazole ring), 1046 (C–S–C); ^1^H NMR (400 MHz, DMSO *d_6_*, δ, ppm): 12.29 (s, 1H, N–H), 8.93 (s, 1H, HC=N), 8.55 (d, *J* = 4.0 Hz, 1H, thiadiazole-H), 8.32 (s, 1H, Ar–H), 8.29 (d, *J* = 4.0 Hz, 1H, Ar–H), 7.78–7.83 (m, 1H, Ar–H), 7.55 (d, *J* = 8.0 Hz, 2H, Ar–H), 7.34–7.26 (m, 3H, Ar–H), 4.67 (s, 2H, SCH_2_); ^13^C NMR (101 MHz, DMSO *d_6_*, δ, ppm): 176.55, 163.50 (thiadiazole ring-C), 160.70 (C=N), 156.48, 149.78, 139.19, 138.01, 137.49, 124.91, 124.24, 123.76, 123.21, 122.68, 122.69, 122.37, 114.74, 113.11, 36.26 (SCH_2_); HR-MS (ESI): calcd. for C_17_H_13_N_5_S_2_: [M + Na^+^] 374.0510; found: 374.0509.

(E)-N-(5-((pyridin-3-ylmethyl)thio)-1,3,4-thiadiazol-2-yl)-1-(1H-indol-3-yl)methanimine (**2c**)

Yellow-green needle-shaped crystal; M. p. 207.5–208.4 °C; yield 92%; IR (ν, cm^–1^ KBr): 3436 (N–H), 3055 (Ar–H), 1605 (C=N), 1580, 1479, 1431, 1294,1241 (thiadiazole ring), 1059 (C–S–C); ^1^H NMR (400 MHz, DMSO *d_6_*, δ, ppm): 12.32 (s, 1H, N–H), 8.92 (s, 1H, HC=N), 8.66 (s, 1H, Ar–H), 8.49 (d, J = 4.0 Hz, 1H, thiadiazole-H), 8.31 (s, 1H, Ar–H), 8.29 (d, J = 4.0 Hz, 1H, Ar–H), 7.88 (d, J = 8.0 Hz, 1H, Ar–H), 7.55 (d, J = 8.0 Hz, 1H, Ar–H), 7.37–7.40 (m, 1H, Ar–H), 7.29 (m, 2H, Ar–H), 4.59 (s, 2H, SCH_2_); ^13^C NMR (101 MHz, DMSO *d_6_*, δ, ppm): 176.75, 163.59 (thiadiazole ring-C), 159.83 (C=N), 150.48, 149.15, 139.27, 138.02, 137.11, 133.44, 124.92, 124.26, 124.09, 122.71, 122.37, 114.75, 113.12, 34.82 (SCH_2_); HR-MS (ESI): calcd. for C_17_H_13_N_5_S_2_: [M + Na^+^] 374.0510; found: 374.051.

(E)-N-(5-((2,4,5-trifluorobenzyl)thiol)-1,3,4-thiadiazol-2-yl)-1-(1H-indol-3-yl)metha- nimine (**2d**)

Bright yellow needle-shaped crystal; M.p. 206.2–207.5 °C; yield 73%; IR (ν, cm^–1^ KBr): 3277 (N–H), 3091(Ar–H), 1620 (C=N),1519, 1423, 1401, 1320, 1239 (thiadiazole ring), 1065 (C–S–C); ^1^H NMR (400 MHz, DMSO *d_6_*, δ, ppm): 12.32 (s, 1H, N–H), 8.94 (s, 1H, HC=N), 8.32 (s, 1H, Ar–H), 8.28 (d, J = 8.0 Hz, 1H, Ar–H), 7.60–7.68 (m, 2H, Ar–H), 7.54 (d, J = 4.0 Hz, 1H, Ar–H), 7.26–7.32 (m, 2H, Ar–H), 4.54 (s, 2H,-SCH_2_); ^13^C NMR (101 MHz, DMSO *d_6_*, δ, ppm): 177.12, 163.70 (thiadiazole ring-C), 159.12 (C=N), 153.98, 139.39, 138.04, 136.79, 128.86, 127.49, 124.92, 124.27, 122.72, 122.37, 119.75, 119.70, 119.55, 114.75, 113.14, 30.81 (SCH_2_); HR-MS (ESI): calcd. for C_18_H_11_F_3_N_4_S_2_: [M + Na^+^] 427.0275; found: 427.0276.

(E)-N-(5-((4-chlorobenzyl)thiol)-1,3,4-thiadiazol-2-yl)-1-(1H-indol-3-yl)methanimine (**2e**)

Beige needle-shaped crystal; M. p. 201.8–202.6 °C; yield 65%; IR (ν, cm^–1^ KBr): 3332 (N–H), 3085 (Ar–H), 1616 (C=N), 1513, 1453, 1428, 1373, 1292 (thiadiazole ring), 1035 (C–S–C); ^1^H NMR (400 MHz, DMSO *d_6_*, δ, ppm): 12.14 (s, 1H, N–H), 9.94 (s, 1H, HC=N), 8.29 (s, 1H, thiadiazole-H), 8.10 (d, *J* = 8.0 Hz, 1H, Ar–H), 7.52 (d, *J* = 8.0 Hz, 1H, Ar–H), 7.38 (d, *J* = 4.0 Hz, 3H, Ar–H), 7.31 (s, 1H, Ar–H), 7.22–7.27 (m, 2H, Ar–H), 4.29 (s, 2H, SCH_2_); ^13^C NMR (101 MHz, DMSO *d_6_*, δ, ppm): 170.49, 163.55 (thiadiazole ring-C), 160.17 (C=N), 149.51, 138.00, 136.88, 136.40, 132.50, 128.89, 124.90, 124.28, 122.72, 122.37, 114.74, 113.11, 38.09 (SCH_2_); HR-MS (ESI): calcd. for C_18_H_13_ClN_4_S_2_: [M + Na^+^] 407.0168; found: 407.0167.

(E)-N-(5-(((1H-benzo[d]imidazol-2-yl)methyl)thiol)-1,3,4-thiadiazol-2-yl)-1-(1H-indol-3-yl)methanimine (**2f**)

Brown-red needle-shaped crystals; M. p. 245.8–246.5 °C; yield 72%; IR (ν, cm^–1^ KBr): 3307 (N–H), 3073 (Ar–H), 1634 (C=N), 1585, 1504, 1454, 1315, 1298 (thiadiazole ring), 1036 (C–S–C); ^1^H NMR (400 MHz, DMSO *d_6_*, δ, ppm): 12.12 (s, 1H, N–H), 10.68 (s, 1H, N–H), 8.39 (s, 1H, HC=N), 8.28 (d, *J* = 8.0 Hz, 1H, Ar–H), 8.17 (s, 1H, Ar–H), 7.98 (d, *J* = 8.0 Hz, 1H, Ar–H), 7.80 (s, 1H, Ar–H), 7.75 (d, *J* = 8.0 Hz, 1H, Ar–H), 7.53 (d, *J* = 8.0 Hz, 1H, Ar–H), 7.38–7.45 (m, 2H, Ar–H), 7.20–7.27 (m, 2H, Ar–H), 4.36 (s, 1H, SCH_2_); ^13^C NMR (101 MHz, DMSO *d_6_*, δ, ppm): 176.03, 170.08 (thiadiazole ring-C), 152.80 (C=N), 149.29, 136.58, 130.17, 129.65, 127.70, 126.98, 123.48, 123.09, 121.30, 118.99, 118.74, 118.61, 114.71, 112.78, 112.03, 111.63, 19.02 (SCH_2_); HR-MS (ESI): calcd. for C_19_H_14_N_6_S_2_: [M–H^+^]: 389.0683; found: 389.070.

(E)-N-(5-((2,6-difluorobenzyl)thio)-1,3,4-thiadiazol-2-yl)-1-(1H-indol-3-yl)methanimine (**2g**)

Light yellow solid powder; M. p. 177.0–177.7 °C; yield 72%; IR (ν, cm^–1^ KBr): 3287 (N–H), 3065 (Ar–H), 1622 (C=N), 1580, 1496, 1409, 1384, 1246 (thiadiazole ring), 1045 (C–S–C); ^1^H NMR (400 MHz, DMSO *d_6_*, δ, ppm): 11.35 (s, 1H, N–H), 9.08 (s, 1H, HC=N), 8.48 (s, 1H, Ar–H), 8.28 (d, J = 4.0 Hz, 1H, Ar–H), 7.58–7.61 (m, 1H, Ar–H), 7.43–7.48 (m, 1H, Ar–H), 7.30–7.34 (m, 2H, Ar–H), 7.09 (t, J = 8.0 Hz, 2H, Ar–H), 4.63 (s, 2H, SCH_2_); ^13^C NMR (101 MHz, DMSO *d_6_*, δ, ppm): 176.99, 162.17 (thiadiazole ring-C), 158.82 (C=N), 137.89, 137.57, 130.42, 124.99, 123.96, 122.43, 122.27, 121.33, 115.24, 112.31, 111.70, 111.45, 25.48 (SCH_2_); HR-MS (ESI): calcd. for C_18_H_12_F_2_N_4_S_2_: [M + Na^+^] 409.0369; found: 409.0369.

(E)-N-(5-(((2-chlorothiazol-5-yl)methyl)thio)-1,3,4-thiadiazol-2-yl)-1-(1H-in-dol-3-yl)me- thanimine (**2h**)

Yellow solid powder; M. p.167.6–169.2 °C; yield 65%; IR (ν, cm^–1^ KBr): 3267 (N–H), 3084 (Ar–H), 1636 (C=N), 1577, 1504, 1462, 1325, 1297 (thiadiazole ring), 1045 (C–S–C); ^1^H NMR (400 MHz, DMSO *d_6_*, δ, ppm): 12.31 (s, 1H, N–H), 8.95 (s, 1H, HC=N), 8.33 (s, 1H, thiadiazole–H), 8.29 (d, J = 4.0 Hz, 1H, Ar–H), 7.66 (s, 1H, Ar–H), 7.55 (d, J = 8.0 Hz, 1H, Ar–H), 7.30 (t, J = 4.0 Hz, J = 8.0 Hz, 2H, thiadiazole–H), 4.81 (s, 2H, SCH_2_); ^13^C NMR (101 MHz, DMSO *d_6_*, δ, ppm): 177.12, 163.74 (thiadiazole ring-C), 162.28 (C=N), 159.30, 150.89, 141.44, 139.37, 138.69, 138.02, 124.92, 124.28,122.74, 122.38, 114.76, 113.13, 29.51 (SCH_2_); HR-MS (ESI): calcd. for C_15_H_10_ClN_5_S_3_: [M + Na^+^] 413.9685; found: 413.96824.

(E)-N-(5-((2,4-dichlorobenzyl)thio)-1,3,4-thiadiazol-2-yl)-1-(1H-indol-3-yl)methanimine (**2i**)

Yellow solid powder; M. p. 182.9–183.8 °C; yield 68%; IR (ν, cm^–1^ KBr): 3273 (N–H), 3093 (Ar–H), 1633 (C=N), 1572, 1504, 1426, 1325, 1238 (thiadiazole ring), 1045 (C–S–C); ^1^H NMR (400 MHz, DMSO *d_6_*, δ, ppm): 12.31 (s, 1H, N–H), 8.93 (s, 1H, HC=N), 8.32 (s, 1H, Ar–H), 8.29 (d, J = 8.0 Hz, 1H, Ar–H), 7.68 (d, J = 4.0 Hz, 1H, Ar–H), 7.61 (d, J = 8.0 Hz, 1H, Ar–H), 7.54 (d, J = 4.0 Hz, 1H, Ar–H), 7.45–7.42 (m, 1H, Ar–H), 7.31–7.261 (m, 2H, Ar–H), 4.62 (s, 2H, SCH_2_); ^13^C NMR (101 MHz, DMSO *d_6_*, δ, ppm): 177.03, 163.68 (thiadiazole ring-C), 159.29 (C=N), 139.33, 138.02, 134.82, 133.88, 133.82, 133.23, 129.38, 128.05, 124.91, 124.27, 122.72, 122.37, 114.75, 113.12, 35.32 (SCH_2_); HR-MS (ESI): calcd. for C_18_H_12_Cl_2_N_4_S_2_: [M–H^+^] 416.98022; found: 416.9815

(E)-N-(5-((4-nitrobenzyl)thio)-1,3,4-thiadiazol-2-yl)-1-(1H-indol-3-yl)methanimine (**2j**)

Brown solid powder; M. p. 194.8–195.5 °C; yield 75%; IR (ν, cm^–1^ KBr): 3433 (N–H), 3042 (Ar–H), 1698 (C=N), 1580, 1518, 1443, 1345,1244 (thiadiazole ring), 1087 (C–S–C); ^1^H NMR (400 MHz, DMSO *d_6_*, δ, ppm): 12.29 (s, 1H, N–H), 8.91 (s, 1H, HC=N), 8.28 (t, J = 12.0 Hz, J = 8.0 Hz, 2H, Ar–H), 8.22 (d, J = 8.0 Hz, 2H, Ar–H), 7.75 (d, J = 8.0 Hz, 2H, Ar–H), 7.54 (d, J = 8.0 Hz, 1H, Ar–H), 7.30–7.27 (m, 2H, Ar–H), 4.70 (s, 2H, SCH_2_); ^13^C NMR (101 MHz, DMSO *d_6_*, δ, ppm): 176.80, 163.55 (thiadiazole ring-C), 159.61(C=N), 147.26, 145.66, 139.26, 138.02, 132.52, 130.80, 124.91, 124.24, 124.11, 122.69, 122.26, 114.74, 113.09, 36.72 (SCH_2_); HR-MS (ESI): calcd. for C_18_H_13_N_5_O_2_S_2_: [M–H^+^] 394.0472; found: 394.0478.

(E)-N-(5-(allylthio)-1,3,4-thiadiazol-2-yl)-1-(1H-indol-3-yl) methanimine (**2k**)

Reddish-brown powder; M. p. 190.3–190.7 °C; yield 72%; IR (ν, cm^–1^ KBr): 3313 (N–H), 3097 (Ar–H), 1639 (C=N), 1574, 1521, 1445, 1392, 1297 (thiadiazole ring), 1047 (C–S–C); ^1^H NMR (400 MHz, Acetone *d_6_*, δ, ppm): 11.20 (s, 1H, N–H), 10.05 (s, 1H, HC=N), 8.23 (t, J = 8.0 Hz, 2H, Ar–H), 7.56 (d, J = 4.0 Hz, 1H, Ar–H), 7.25–7.29 (m, 1H, Ar–H), 6.64 (s, 1H, =CH), 5.91–6.01 (m, 1H, =CH), 5.25 (d, J = 12.0 Hz, 1H, =CH), 5.12 (d, J = 8.0 Hz, 1H, =CH), 3.75 (d, J = 8.0 Hz, 2H, SCH_2_); ^13^C NMR (101 MHz, Acetone *d_6_*, δ, ppm): 181.65, 170.07 (thiadiazole ring-C), 162.28 (C=N), 150.68, 137.51, 133.41, 124.66, 123.58, 122.15, 121.31, 119.06, 118.63, 118.06, 112.31, 37.48 (SCH_2_); HR-MS (ESI): calcd. for C_14_H_12_N_4_S_2_: [M + Na^+^] 323.0401; found: 323.0401.

(E)-N-(5-((1-phenylallyl)thiol)-1,3,4-thiadiazol-2-yl)-1-(1H-indol-3-yl)methanimine (**2l**)

Yellow solid powder; M. p. 217.0–218.3 °C; yield 69%; IR (ν, cm^–1^ KBr): 3470 (N–H), 3040 (Ar–H), 1642 (C=N), 1574, 1510, 1457, 1373, 1241 (thiadiazole ring), 1064 (C–S–C); ^1^H NMR (400 MHz, DMSO *d_6_*, δ, ppm): 12.29 (s, 1H, N–H), 8.92 (s, 1H, HC=N), 8.31 (s, 1H, Ar–H), 8.29 (d, J = 8.0 Hz, 1H, Ar–H), 7.54 (d, J = 8.0 Hz, 1H, Ar–H), 7.44 (s, 3H, Ar–H), 7.29 (d, J = 4.0 Hz, 2H, Ar–H), 6.72 (q, J = 12.0 Hz, J = 8.0 Hz, J = 12.0 Hz, 1H, Ar–H), 5.83 (d, J = 20.0 Hz, 1H, Ar–H), 5.26 (d, J = 12.0 Hz, 1H, =CH), 4.55 (s, 2H, =CH_2_), 4.37 (t, J = 4.0 Hz, 1H, SCH); ^13^C NMR (101 MHz, DMSO *d_6_*, δ, ppm): 176.53, 163.46 (thiadiazole ring-C), 160.42 (C=N), 139.19, 138.01, 136.95, 136.77, 136.64, 129.84, 126.79, 124.92, 122.69, 122.37, 115.02, 114.74, 113.10, 37.62 (SCH_2_); HR-MS (ESI): calcd. for C_20_H_16_N_4_S_2_: [M + Na^+^] 399.0714; found: 399.0714.

(E)-N-(5-((pyridin-4-ylmethyl)thiol)-1,3,4-thiadiazol-2-yl)-1-(1H-indol-3-yl)methanimine (**2m**)

Orange solid powder; M. p. 183.2–184.5 °C; yield 65%; IR (ν, cm^–1^ KBr): 3442 (N–H), 3043 (Ar–H), 1633 (C=N), 1577, 1521, 1445, 1396, 1244 (thiadiazole ring), 1087 (C–S–C); ^1^H NMR (400 MHz, DMSO *d_6_*, δ, ppm): 12.32 (s, 1H, N–H), 8.92 (s, 1H, HC=N), 8.66 (s, 1H, Ar–H), 8.49 (d, J = 8.0 Hz, 1H, Ar–H), 8.31 (s, 1H, Ar–H), 8.28 (d, J = 4.0 Hz, 1H, Ar–H), 7.88 (d, J = 8.0 Hz, 1H, Ar–H), 7.55 (d, J = 8.0 Hz, 1H, Ar–H), 7.38 (q, J = 4.0 Hz, 1H, Ar–H), 7.26–7.32 (m, 2H, Ar–H), 4.59 (s, 2H, SCH_2_); ^13^C NMR (101 MHz, DMSO *d_6_*, δ, ppm): 172.6, 169.31 (thiadiazole ring-C), 165.56 (C=N), 159.22, 156.40, 155.03, 137.40, 137.16, 124.63, 123.61, 122.12, 121.33, 119.21, 112.09, 35.79 (SCH_2_); HR-MS (ESI): calcd. for C_17_H_13_N_5_S_2_: [M + H^+^] 352.0691; found: 352.0691

(E)-N-(5-((3-bromo-2-fluorobenzyl)thio)-1,3,4-thiadiazol-2-yl)-1-(1H-indol-3-yl)meth- animine (**2n**)

Yellow solid powder; M. p. 198.0–199.3 °C; yield 76%; IR (ν, cm^–1^ KBr): 3439 (N–H), 3053 (Ar–H), 1605 (C=N), 1577, 1482, 1459, 1392, 1241 (thiadiazole ring), 1053 (C–S–C); ^1^H NMR (400 MHz, DMSO *d_6_*, δ, ppm): 11.36 (s, 1H, N–H), 9.05 (s, 1H, HC=N), 8.46 (d, *J* = 8.0 Hz, 1H, Ar–H), 8.27(s, 1H, Ar–H), 7.55–7.60 (m, 2H, Ar–H), 7.42–7.47 (m, 2H, Ar–H), 7.31–7.34 (m, 2H, Ar–H), 4.60 (s, 2H, SCH_2_); ^13^C NMR (101 MHz, DMSO *d_6_*, δ, ppm): 177.03, 163.68 (thiadiazole ring-C), 159.29 (C=N), 138.02, 134.82, 133.88, 133.82, 133.23, 129.58, 128.05, 124.91, 124.27, 122.72, 122.37, 114.75, 113.12, 35.32 (SCH_2_); HR-MS (ESI): calcd. for C_18_H_12_BrFN_4_S_2_: [M + Na^+^] 468.9569; found: 468.9575.

(E)-N-(5-((3-methoxybenzyl)thio)-1,3,4-thiadiazol-2-yl)-1-(1H-indol-3-yl)methanimine(**2o**)

Yellow solid powder; M. p. 194.2–195.4 °C; yield 73%; IR (ν, cm^–1^ KBr): 3464 (N–H), 3064 (Ar–H), 1636 (C=N), 1577, 1462, 1440, 1389, 1244 (thiadiazole ring), 1050 (C–S–C); ^1^H NMR (400 MHz, DMSO *d_6_*, δ, ppm): 11.18 (s, 1H, N–H), 8.9 (s, 1H, HC=N), 8.31 (d, *J* = 8.0 Hz, 1H, Ar–H), 8.06–8.12 (m, 2H, Ar–H), 7.44 (d, 1H, *J* = 8.0 Hz, Ar–H), 7.12–7.19 (m, 3H, Ar–H), 6.94 (t, *J* = 8.0 Hz, 2H, Ar–H), 4.42 (s, 2H, SCH_2_), 3.66 (s, 3H, OCH_3_); ^13^C NMR (101 MHz, DMSO *d_6_*, δ, ppm): 176.07, 161.88 (thiadiazole ring-C), 159.98 (C=N), 138.28, 137.86, 137.32, 129.62, 129.50, 123.92, 123.61, 122.42, 122.22, 122.11, 121.31, 115.24, 114.71, 113.20, 112.28, 54.63 (OCH_3_), 37.52 (SCH_2_); HR-MS (ESI): calcd. for C_19_H_16_N_4_OS_2_: [M + Na^+^] 403.0663; found: 403.0667.

(E)-N-(5-((3-(trifluoromethyl)benzyl)thiol)-1,3,4-thiadiazol-2-yl)-1-(1H-indol-3-yl)meth- animine (**2p**)

Yellow solid powder; M. p. 201.3–202.2 °C; yield 68%; IR (ν, cm^–1^ KBr): 3419 (N–H), 3069 (Ar–H), 1670 (C=N), 1577, 1462, 1426, 1328, 1246 (thiadiazole ring), 1064 (C–S–C); ^1^H NMR (400 MHz, Acetone *d_6_*, δ, ppm): 11.34 (s, 1H, N–H), 9.06 (d, *J* = 20.0 Hz, 1H, HC=N), 8.45 (d, *J* = 20.0 Hz, 1H, Ar–H), 8.27 (d, *J* = 8.0 Hz, 1H, Ar–H), 7.4 (q, *J* = 8.0 Hz, *J* = 12.0 Hz, 2H, Ar–H), 7.59 (d, *J* = 8.0 Hz, 1H, Ar–H), 7.20–7.46 (m, 4H, Ar–H), 4.70 (d, *J* = 12.0 Hz, 2H, SCH_2_); ^13^C NMR (101 MHz, Acetone *d_6_*, δ, ppm): 176.31, 162.05 (thiadiazole ring-C), 159.67 (C=N), 142.10, 137.87, 137.44, 129.91, 125.39, 123.95, 122.41, 122.25, 115.23, 112.31, 36.54 (SCH_2_); HR-MS (ESI): calcd. for C_19_H_13_F_3_N_4_S_2_: [M + Na^+^] 441.0431found: 441.0431.

(E)-N-(5-((3,5-bis(trifluoromethyl)benzyl)thiol)-1,3,4-thiadiazol-2-yl)-1-(1H-indol-3-yl)me -thanimine (**2q**)

Yellow solid powder; M. p. 201.3–202.2 °C; yield 72%; IR (ν, cm^–1^ KBr): 3456 (N–H), 3066 (Ar–H), 1650 (C=N), 1577, 1496, 1437, 1375, 1243 (thiadiazole ring), 1050 (C–S–C); ^1^H NMR (400 MHz, Acetone *d_6_*, δ, ppm): 11.35 (s, 1H, N–H), 9.03 (s, 1H, HC=N), 8.45 (d, *J* = 8.0 Hz, 1H, Ar–H), 8.26 (d, *J* = 4.0 Hz, 1H, Ar–H), 8.24 (s, 2H, Ar–H), 7.99 (s, 1H, Ar–H), 7.59 (t, *J* = 4.0 Hz, 1H, Ar–H), 7.29–7.35 (m, 2H, Ar–H), 4.83 (s, 2H, SCH_2_); ^13^C NMR (101 MHz, Acetone *d_6_*, δ, ppm): 176.58, 162.18 (thiadiazole ring-C), 159.13 (C=N), 141.31, 137.88, 137.56, 131.36, 131.04, 129.99, 124.97, 123.96, 122.41, 122.28, 121.23, 115.21, 112.32, 35.79 (SCH_2_); HR-MS (ESI): calcd. for C_20_H_12_F_6_N_4_S_2_: [M–H^+^] 487.0486; found: 487.0486.

(E)-N-(5-((2-chloro-6-fluorobenzyl)thiol)-1,3,4-thiadiazol-2-yl)-1-(1H-indol-3-yl)meth- animine (**2r**)

Yellow solid powder; M. p. 194.6–195.5 °C; yield 63%; IR(ν, cm^–1^ KBr): 3489 (N–H), 3063 (Ar–H), 1622 (C=N), 1577, 1493, 1431, 1381, 1243 (thiadiazole ring), 1061 (C–S–C); ^1^H NMR (400 MHz, Acetone *d_6_*, δ, ppm): 11.34 (s, 1H, N–H), 9.09 (s, 1H, HC=N), 8.47 (d, *J* = 8.0 Hz, 1H, Ar–H), 8.28 (d, *J* = 4.0 Hz, 1H, Ar–H), 7.60 (t, *J* = 4.0 Hz, 1H, Ar–H), 7.39–7.45 (m, 2H, Ar–H), 7.35–7.32 (m, 2H, Ar–H), 7.22 (t, *J* = 8.0 Hz, 1H, Ar–H), 4.73 (s, 2H, SCH_2_); ^13^C NMR (101 MHz, Acetone *d_6_*, δ, ppm): 177.02, 162.14 (thiadiazole ring-C), 160.16 (C=N), 158.79, 137.88, 137.54, 130.56, 130.46, 125.75, 123.96, 122.43, 122.27, 117.01, 115.26, 114.63, 114.41, 112.30, 37.18 (SCH_2_); HR-MS (ESI): calcd. for C_18_H_12_ClFN_4_S_2_: [M + Na^+^] 425.0074; found: 425.0071.

(E)-N-(5-((4-(trifluoromethyl)benzyl)thiol)-1,3,4-thiadiazol-2-yl)-1-(1H-indol-3-yl)meth- animine (**2s**)

Yellow solid powder; M. p. 198.1–199.0 °C; yield 65%; IR(ν, cm^–1^ KBr): 3442 (N–H), 3043 (Ar–H), 1653 (C=N), 1574, 1490, 1442, 1389, 1243 (thiadiazole ring), 1081 (C–S–C); ^1^H NMR (400 MHz, Acetone *d_6_*, δ, ppm): 12.29 (s, 1H, N–H), 8.91 (s, 1H, HC=N), 8.28 (t, *J* = 12.0 Hz, *J* = 8.0 Hz, 2H, Ar–H), 8.22 (d, *J* = 8.0 Hz, 2H, Ar–H), 7.75 (d, *J* = 8.0 Hz, 2H, Ar–H), 7.54 (d, *J* = 8.0 Hz, 1H, Ar–H), 7.28 (s, 2H, Ar–H), 4.70 (s, 2H, SCH_2_); ^13^C NMR (101 MHz, Acetone *d_6_*, δ, ppm): 176.31, 162.05 (thiadiazole ring-C), 159.67 (C=N), 142.10, 137.87, 137.44, 129.91, 128.91, 125.42, 124.99, 123.95, 122.41, 122.25, 115.23, 112.31, 36.54 (SCH_2_); HR-MS (ESI): calcd. for C_19_H_13_F_3_N_4_S_2_: [M–H^+^] 417.0456; found: 417.0496.

(E)-N-(5-((2-(trifluoromethyl)benzyl)thio)-1,3,4-thiadiazol-2-yl)-1-(1H-indol-3-yl)meth- animine (**2t**)

Yellow solid powder; M. p. 194.7–195.6 °C; yield 63%; IR (ν, cm^–1^KBr): 3444 (N–H), 3043 (Ar–H), 1636 (C=N), 1561, 1496, 1448, 1336, 1246 (thiadiazole ring), 1055 (C–S–C); ^1^H NMR (400 MHz, Acetone *d_6_*, δ, ppm): 12.34 (s, 1H, N–H), 9.03 (d, *J* = 20.0 Hz, 1H, HC=N), 8.46 (t, *J* = 8.0 Hz, 1H, Ar–H), 8.27 (d, *J* = 8.0 Hz, 1H, Ar–H), 7.58–7.78 (m, 4H, Ar–H), 7.23–7.39 (m, 3H, Ar–H), 4.71 (d, *J* = 8.0 Hz, 2H, SCH_2_); ^13^C NMR (101 MHz, Acetone *d_6_*, δ, ppm) δ: 176.31, 162.05 (thiadiazole ring-C), 159.67 (C=N), 142.10, 142.10, 137.87, 137.44, 129.91, 128.94, 125.76, 125.42, 124.99, 123.95, 123.06, 122.41, 122.25, 115.23, 112.31, 36.54 (SCH_2_); HR-MS (ESI): calcd. for C_19_H_13_F_3_N_4_S_2_: [M–H^+^] 417.0456; found: 417.0496.

Spectra for structural information about the compounds are provided in the Appendix A.

### 3.4. In Vitro Antifungal Assay

The antifungal activities of the novel compounds **2a**–**2t** were tested based on the reported method [74]. The synthesized compounds were dissolved in a 20% acetone water solution. The solution of each compound was added to sterilized potato dextrose agar to obtain a final concentration of 500 μg/mL. After the mixture was cooled, the mycelium of the fungi was transferred to the test plate and incubated at 25 °C for 4–7 days. When the mycelium reached the edges of the control plate (without the added samples), the inhibitory index was calculated using the following formula:Inhibitory index (%) = (1 − Da/Db)
where Da is the diameter of the growth zone in the test plate, and Db is the diameter of the growth zone in the control plate. Each experiment was performed three times and the data points were averaged. The commercial fungicide triadimefon (100 μg/mL) was used as a control and tested in the same manner.

## 4. Conclusions

In the present study, a series of novel indole derivatives containing 1,3,4-thiadiazole scaffolds modified with thioether groups were efficiently designed and synthesized. In addition, their antifungal activities were investigated against *F. graminearum, F. oxysporum*, *F.*
*moniliforme, C. lunata*, and *P. p.* var. *nicotianae*. The antifungal activity test results show that some of the indole analogs exhibited better antifungal activity than the control reagent triadimefon. Compound **2j** was identified as the most active against *F. graminearum*, *F. oxysporum*, *F.*
*moniliforme,* and *P. p.* var. *nicotianae* with the inhibition rates of 100%, 95.7%, 89%, and 76.5%, respectively. Compounds **2j** and **2q** exhibited better antifungal activity against *C. lunata* with inhibition rates of 81.9% and 83.7%, respectively. Compound **2j,** as the representative compound, was used for further mechanistic studies. The indole derivatives containing modified 1,3,4-thiadiazole with the electron-withdrawing –NO_2_ group on the benzene ring showed better antifungal activity. Conclusively, the structural optimization of indole derivatives containing modified 1,3,4-thiadiazole with the electron-withdrawing groups on the benzene ring is a potential strategy to prepare analogs with improved antifungal activity.

## 5. Patents

There is a patent resulting from the work reported in this manuscript.

## Data Availability

The data presented in this study are available on request from the corresponding author.

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
