# Peer review of "Synthesis of Novel Indole Schiff Base Compounds and Their Antifungal Activities"

_molecules, 2022, doi:10.3390/molecules27206858_

Round 1
Reviewer 1 Report
Authors present a series of novel indole Schiff base derivatives containing a 1,3,4-thiadiazole scaffold modified with a thioether group. The structures of these compounds were confirmed by FT-IR, NMR, HR-MS and antifungal activity was investigated.
Introduction contains motivations of authors to research new tiadiazole derivative with indole moiety which can be used as potential fungicides. All methods are clearly describe and results of experiment are presents. I think that the conclusions of the substituent's influence on biological activity should be more described. Will the nitro group in another position (orto or meta) have the same properties?
In text I find some small mistakes:
- in „Affiliation” should be name of university
- arrow on scheme 1 looks like a retrosynthetic arrow and I think that a simple arrow is more suitable.
Reviewer 2 Report
Dear Authors,
You present here the chemical synthesis of some new Schiff bases containing 1,3,4 thiadiazole and their antifungal investigation.
The subject is interesting. Still, I have some suggestions:
- in lines 10-12, in the Abstract part, you don't need to explain in details the name of the techniques used. Plus, the abbreviations are world-widely known, such as NMR
- what do you consider the difference between the "anti-tumor" and the "anti-cancer" activities?
- you should correct "1,3,4 triadiazole" into "1,3,4 thiadiazole", in line 64
- in Figure 1, you should correct "anaigesic" with "analgesic"
- for the numbers of compounds in Figures 1 and 2, there is a different font used
- in Figure 2, correct "thiadiazole"
- draw compound V in order to be clear
- the schemes should be presented after they have been cited in text
- correct "thiodiazole" in line 119
- the figures should look alike
- I suggest you present all results obtained and only after, make the discussion regarding the chemical structure-antifungal activity relationship, in order to avoid repetition
- correct in the compounds' characterization thiadiazole, not "thiodiazole", not "thia-diazole"
- the chemical synthesis is a very simple one, following a very well-known protocol, so there is a total lack of innovation and novelty in the chemical part
- you need to do a toxicity test, in order to provide information regarding the safety of the compounds' use
Reviewer 3 Report
I have reviewed manuscript number": molecules-1951541, entitled “Synthesis of Novel Indole Schiff Base Compounds and Their Antifungal Activities”. The authors prepared novel indole Schiff base compounds. They characterize the compounds by FT-IR, 1H & 13C NMR, and HR-MS. They also explore the antifungal activities of the compounds. The data support the preparation of the compounds, although the lack of x-ray analysis of complexes is a negative point. Overall, the manuscript looks good but significant amount of improvement is needed in the introduction part and other part of the manuscript before further consideration. So, I am recommending minor revision. The comments are:
- The main objectives and its novelty should be summarized.
2. In the introduction, the authors have written: “Schiff base, a class of compounds with imine groups (-CH=N-), is a common pharmacological group in many compounds, which has a wide range of antifungal, antibacterial and other biological activities.….”. The authors must add the following references:
(antibacterial) : https://doi.org/10.1007/s13738-018-1347-6
(antifungal) :https://doi.org/10.1080/00958972.2021.1990271
- Authors should comment on the solubility nature of the synthetic compounds in the revised manuscript.
- Whether the synthetic molecules stable in solution? They should comment on the molecular stability in the solution phase.
- Some “References” should be corrected based on the “Molecules” format.
- The English language of the manuscript should be carefully checked and some typos should be corrected
Round 2
Reviewer 2 Report
Dear Authors,
Thank you for considering my suggestions and that you made the changes that I told you. Keep the work in this field and do not neglect the importance of the toxicity investigation of the synthesized compounds.